# Mouse Model of STAT3 Mutation Resulting in Job’s Syndrome Diverges from Human Pathology

**DOI:** 10.3390/ijms26167675

**Published:** 2025-08-08

**Authors:** Jakub Jankowski, Jichun Chen, Gyuhyeok Cho, Sung-Gwon Lee, Chengyu Liu, Neal Young, Jungwook Kim, Lothar Hennighausen

**Affiliations:** 1Section of Genetics and Physiology, Laboratory of Cellular and Molecular Biology, National Institute of Diabetes and Digestive and Kidney Diseases, US National Institutes of Health, Bethesda, MD 20892, USA; sunggwonl22@gmail.com (S.-G.L.); lotharh@niddk.nih.gov (L.H.); 2Hematology Branch, National Heart, Lung, and Blood Institute, National Institutes of Health, Bethesda, MD 20892, USA; chenji@nhlbi.nih.gov (J.C.); youngns@nhlbi.nih.gov (N.Y.); 3Department of Chemistry, Gwangju Institute of Science and Technology, Gwangju 61005, Republic of Korea; gyuhyeokcho@gm.gist.ac.kr (G.C.); jwkim@gist.ac.kr (J.K.); 4Transgenic Core, National Heart, Lung, and Blood Institute, US National Institutes of Health, Bethesda, MD 20892, USA; liuch@nhlbi.nih.gov

**Keywords:** STAT3, hyper-IgE syndrome, in-frame deletion

## Abstract

*STAT3* mutations are commonly observed in human pathology yet have no uniform patient presentation. Their effects range from cancer and autoimmunity to primary immunodeficiencies and bone deformity. Designing animal models of those mutations can help researchers identify their direct effects to better inform the clinical setting. In this manuscript, we report a mouse model harboring the same mutation as an autosomal-dominant hyper-IgE syndrome (AD-HIES) patient reported in the literature. Surprisingly, while the deletion of five amino acids in the SH2 domain of STAT3 did result in frequency changes in several immune populations as measured by complete blood count and flow cytometry analysis, it did not yield the expected phenotype of AD-HIES, with no increase in serum IgE or eosinophil count. We additionally provide structural analysis of the *STAT3^G656_M660del^* deletion, visualizing changes in protein architecture and potential effects on the neighboring Y705 phosphorylation site. Our model showcases the sexually dimorphic immune dysregulation caused by a *STAT3* mutation and highlights that predicted gain- and loss-of-function mutations can yield unexpected phenotypes.

## 1. Introduction

STAT3 is a backbone of immune response, directing lymphocyte differentiation, cytokine release and cell death [1,2]. It is not surprising that its mutations often have severe consequences, and animal models of homozygous knockout are not viable [3]. Human pathology of *STAT3* mutations encompasses numerous chronic conditions, ranging from autoimmunity to primary immunodeficiency. Those effects are often divided between gain- and loss-of-function STAT3 mutations, with the overactive STAT3 contributing to immune response overdrive, and its lower activity to immunosuppression, a mechanism both observed in pathology and used in the clinic [2,4,5,6]. This division, however, has recently become blurrier, as the number of patients harboring identified *STAT3* mutations and descriptions of their pathophysiology increased, showcasing the variability of their clinical presentation [7,8,9].

In this manuscript, we describe a mouse model possessing a five-amino acid in-frame deletion identified in a human hyper-IgE syndrome (HIES) patient [10]. HIES, and more specifically Job’s syndrome, its variant defined by heterozygous loss-of-function *STAT3* mutations, is a subtype of primary immunodeficiency. It results in characteristic facial abnormalities, eczema, and susceptibility to infection. It is usually clinically validated via increased serum IgE levels, often accompanied by eosinophilia and a decrease in Th-17 cells; however, the last two can be evident only due to active infection, when HIES patients are actively seeking medical help and are diagnosed [11,12,13]. While the mouse did not show IgE elevation, its phenotype indicates immune imbalance with an increased number of lymphocytes, monocytes and neutrophils, indicating gain, rather than loss, of STAT3 function and thus warranting further investigation.

## 2. Results

### 2.1. Generation of G656_M660del Mice

*G656_M660del* mice were generated as a result of off-target CRISPR/Cas9 modification aiming to introduce STAT3 D661H SNP into the genome (Appendix A). Instead of the intended effect, 15 nucleotides upstream of the substitution site were removed (656–660); alternatively, due to the identical in-frame sequence, nucleotides 654–658 could have been removed, yielding the same result. While sgRNA used to generate the strain has several possible off-target sites, they possess a minimum of two mismatches and do not occur in gene exons (Appendix A).

### 2.2. HIES Phenotype Investigation

We conducted a literature search for similar deletions to investigate the clinical relevance of our strain and discovered an AD-HIES patient harboring the same mutation [10]. She presented with eczema, retained primary teeth, joint hyperflexibility and recurrent infections. As the mice showed no skin abnormalities or any other visible phenotype change, we measured IgE levels to confirm HIES phenotype (Figure 1a). We used heterozygous *G656_M660del* mice, as Job’s syndrome is an autosomal-dominant disease; additionally, our breeding attempts did not yield homozygous mice despite both heterozygous females and males being fertile. STAT3 gain-of-function mutations and deficiency were both observed to be lethal embryonically or shortly after birth [3,14]. Heterozygous breeding up to date yielded six litters, totaling 22 heterozygous and 9 wildtype pups in addition to six non-viable litters we were unable to recover to assess genotype.

*G656_M660del* mice did not have elevated IgE as measured by ELISA (Figure 1a). While we observed a considerable dimorphism between females and males, often noted in the literature, it was not dependent on genotype (*p* < 0.0001 for both males and females) [15,16,17]. Because the assay designer did not account for this effect and the lowest IgE concentrations were at the detection limit of the assay, we repeated the experiment with 10-fold lower serum dilution (Appendix A), which still did not yield a result reflecting HIES.

### 2.3. Effects of the STAT3^G656_M660del^ Deletion on Blood Cell Populations

As described earlier, STAT3 has a prominent role in regulating immune cell differentiation and activity. We investigated whether the deletion has any effect on circulating immune cell populations. We utilized complete blood count (CBC) and flow cytometry to quantify the cells and made a number of observations (Figure 1b–h, Appendix A). First, there was no notable increase or decrease in Il-17A-expressing CD4 T-cells characteristic for STAT3 deregulation (*p* = 0.9998 and *p* = 0.5605 for females and males, respectively). *G656_M660del* males, exhibited elevated counts of total white blood cells and lymphocytes compared to mutant females (*p* = 0.002 and 0.0031, respectively), as well as more neutrophils and monocytes when compared to WT males (*p* < 0.0001 and *p* = 0.0068) in addition to the sexually dimorphic effect. *G656_M660del* mice had significantly less platelets than WT (*p* = 0.0216 and *p* = 0.045 for females and males, respectively) and red blood cells, especially visible in females (*p* < 0.0001 and *p* = 0.0246). Further CBC and flow cytometry investigation revealed no eosinophil increase in mutants and no further changes in immune cell populations, with sexual dimorphism of immune cell frequency intact (Appendix A). This phenotype persisted in 4-month-old mice as shown by additional CBC analyses (Appendix A).

### 2.4. Structural Changes in the G656_M660del Mutant

The deletion of five amino acids (656GYKIM660) occurred within the Src homology 2 (SH2) domain of STAT3 [18]. To investigate the structural consequences of this deletion, we generated a mutant model using AlphaFold3 [19,20]. Based on the crystal structure of phosphorylated wild-type (WT) STAT3β dimer bound to double-stranded DNA (dsDNA) (PDB: 1BG1), we modeled both WT and deletion mutant structures spanning residues 127-722, along with dsDNA (Figure 2a).

Both AF3-predicted structures resembled the WT crystal structure, with root mean square deviation (RMSD) values of 1.338 Å for the WT model and 0.943 Å for the deletion mutant. The deleted region lies near the C-terminus of the SH2 domain, close to the dimerization interface (Figure 2b). In the WT crystal structure, the 656GYKIM660 peptide was located within an extended loop (residues G656-E680) (Figure 2c), stretching in parallel to the C-terminal β-strand. The AF3-predicted WT model also showed a similar parallel extension of the pentapeptide; however, in this model, residues K658-D661 additionally formed a short β-strand (Figure 2d).

Deletion of the pentapeptide shortened the loop and promoted an extension of the adjacent α-helix (Figure 2e). Specifically, the α-helix expanded from six (α-helix650-655, WT crystal) or seven residues (α-helix650-656, WT AF3) to eleven residues (α-helix650-656 and 661-666) in the mutant. The extended α-helix remained parallel to the C-terminal β-strand. Interestingly, the C-terminus of the mutant model formed an additional α-helix (α-helix715-721), whereas the C-termini of both WT structures (crystal and AF3) appeared unstructured.

In the WT structures, the pentapeptide formed stabilizing hydrogen bonds with the C-terminal β-strand (Figure 2f,g). For example, interactions between Y657 and T714, and between M660 and T714, were observed in both the crystal and AF3-predicted WT models. These interactions were lost in the deletion mutant (Figure 2h), weakening the interface between the α-helix650-656 and 661–666 and the C-terminal β-strand. Additionally, deletion introduced more hydrophobic residues (I665 and L666) into the interface, further reducing its stability.

Notably, the deleted region lies close to the critical phosphorylation site Y705 [18]. Thus, the deletion could affect phosphorylation efficiency by altering the local environment around Y705, potentially impacting recognition by JAK family kinases. Furthermore, as the deletion is near the SH2 dimerization interface, it could influence STAT3 dimerization, which is essential for nuclear translocation and transcriptional activation [21].

## 3. Discussion

There are over 150 *STAT3* mutations identified in HIES patients [22]. Despite that, no clear pattern emerges as to why those particular changes cause loss of protein function. While the majority of them are missense mutations and cluster to the DNA-binding and SH2 domains, the mechanism behind Job’s syndrome development remains unclear. Our mouse model, though mimicking a human *STAT3* mutation, did not develop HIES. The current assumption is that loss of STAT3 activity is sufficient for the disease to develop, but the phenotype is highly inconsistent, and the severity and presence of symptoms can vary [12]. This is partially rooted in the difficulty and delay of diagnosis and the resulting widespread age of the investigated patient cohorts [23].

One mouse model was able to partially recapitulate human disease [24]. This *mut-STAT3* strain expressed two additional copies of a common HIES ΔV463 STAT3 variant to equilibrate it with two WT copies of the gene and thus mimicking a heterozygote. The resulting strain displayed increased IgE, lower Il-17 production, and attenuated response to immune system stimulation. This suggests that the variety of underlying mutations might play a role in the severity of symptoms, perhaps by differing in the degree to which protein structure is altered.

Similarity to HIES notwithstanding, our model contributes to continued dissection of STAT3 function and its role in pathophysiology. First, heterozygous *G656_M660del* mutation is enough to affect lymphocyte and myeloid numbers, but only in males. While not present in HIES, similar dimorphism has been observed in other studies, for example, in myocardial inflammation and lung cancer, and estrogen receptor signaling has been implicated as its driver [25,26]. Further, there is an established association between STAT3 activity and anemia [27,28,29]. The decrease in red blood cells and linked readouts is larger in females, and might provide a clue towards understanding its development, as human patients also display sexual dimorphism of anemia susceptibility [30]. Lastly, thrombocytopenia is only vaguely linked to STAT3, though it is a known activator of platelet function [31].

Taken together, those results paint a picture of a mouse model presenting with a likely *STAT3* gain-of-function mutation, despite reflecting a HIES patient’s *STAT3* variant. This finding confirms the 656-661 amino acids of STAT3 as a hotspot for mutations significantly altering the protein’s activity and structure. In conjunction with our other research, we showcase that gain- and loss-of-function labels might not be enough for the nuanced activity of STAT proteins [32,33].

## 4. Materials and Methods

### 4.1. Mice

All animals were housed and handled according to the Guide for the Care and Use of Laboratory Animals (8th edition) and all animal experiments were approved by the Animal Care and Use Committee (ACUC) of National Institute of Diabetes and Digestive and Kidney Diseases (NIDDK, MD) and performed under the NIDDK animal protocol K089-LGP-23. CRISPR/Cas9 targeted mice were generated using C57BL/6N mice (Charles River, Wilmington, MA, USA) by the Transgenic Core of the National Heart, Lung, and Blood Institute (NHLBI) as described previously (Appendix A) [32]. After identifying the introduced mutation, heterozygous founders were bred with WT C57BL/6N mice. Mice from F2 and F3 generations were used in the study. A total of 55 mice were used to obtain the data described, and no exclusion criteria except age- and sex-matching were used. All procedures were performed on 2-month-old animals, except Appendix A (4 months old) and Figure 1b (7 months old).

### 4.2. Serum IgE Measurement

Blood was collected from retro-orbital sinus into Eppendorf tubes in the presence of 5 mM ethylenediaminetetraacetic acid (EDTA, Sigma-Aldrich, St. Louis, MO, USA). Serum IgE levels were measured using ELISA (Cat. EMIGHE, Invitrogen, Carlsbad, CA, USA) according to the manufacturer’s instructions.

### 4.3. Complete Blood Count and Flow Cytometry

Blood was collected from retro-orbital sinus into Eppendorf tubes in the presence of 5 mM ethylenediaminetetraacetic acid (EDTA, Sigma). Complete blood counts (CBCs) were performed using an Element HT5 analyzer (Heska Corporation, Loveland, CO, USA). To perform surface antigen staining, the remaining blood was washed using ACK buffer, and the following antibodies were used to detect surface markers: PE Annexin V (BioLegend 640947, San Diego, CA, USA), Brilliant Violet 421™ anti-mouse F4/80 (BioLegend 123132), Pacific Blue™ anti-mouse CD4 Antibody (BioLegend 100428), FITC anti-mouse CD62L Antibody (BioLegend 104406), PE/Cyanine5 anti-mouse/human CD45R/B220 Antibody (BioLegend 103210), PerCP/Cyanine5.5 anti-mouse/human CD11b Antibody (BioLegend 101228), PE/Cyanine7 anti-mouse CD8a Antibody (BioLegend 100722), APC anti-mouse/human CD44 Antibody (BioLegend 103012), Alexa Fluor^®^ 647 anti-mouse CD95 (Fas) Antibody (BioLegend 152620), and APC/Cyanine7 anti-mouse Ly-6G Antibody (BioLegend 127624). 7AAD was used as the viability dye. Intracellular Il-17A staining was performed by adding 45 min of incubation in fixation/permeabilization buffer (Invitrogen/eBioscience, San Diego, CA, USA) before washing and adding following antibodies: APC/Cyanine7 anti-mouse CD8a Antibody (BioLegend 100714), PE/Cyanine5 anti-mouse/human CD45R/B220 Antibody (BioLegend 103210), APC anit-mouse IL-17A Monoclonal Antibody (ThermoFisher 14-7177-81, Waltham, MA, USA), PE/Cyanine7 anti-mouse CD4 Antibody (BioLegend 100422), and FITC anti-mouse CD3 Antibody (BioLegend 100204). Flow cytometry data was acquired using Cytek Aurora system (Cytek Biosciences, Fremont, CA, USA) and analyzed using SpectroFlo 3.3.0 software (Cytek Biosciences). Representative gating strategies were presented as Appendix A.

### 4.4. Structural Analysis

The protein sequence of STAT3β (residues 127-722) was obtained from UniProt and the double-stranded DNA (dsDNA) sequence was extracted from the Protein Data Bank (PDB ID: 1BG1) [34,35]. Structural models were generated using AlphaFold3 to predict a STAT3β127-722 dimer complexed with dsDNA and phosphorylated at Y705 [19,20]. The resulting structures were aligned and visualized using PyMOL (The PyMOL Molecular Graphics System, Version 3.1; Schrödinger, LLC, New York, NY, USA). All AlphaFold3-predicted structural models used in this study were provided in the Appendix A.

### 4.5. Statistical Analysis

All samples were randomly selected and only limited by genotype availability. Blinding was applied to CBC and flow cytometry analyses. For comparison of samples, data were presented as standard error of the mean. Data was analyzed for outliers (ROUT method, Q = 1%) and normal distribution (Shapiro–Wilk), followed by a 2-way ANOVA with Tukey correction for multiple comparisons or a non-parametric test. F-test was used to compare variance between groups. GraphPad PRISM 10.0.1 software was used for analysis. Values of * *p* < 0.05, ** *p* < 0.01, *** *p* < 0.001, and **** *p* < 0.0001 were considered statistically significant. Group sizes are described in the figure descriptions.

## Figures and Tables

**Figure 1 ijms-26-07675-f001:**
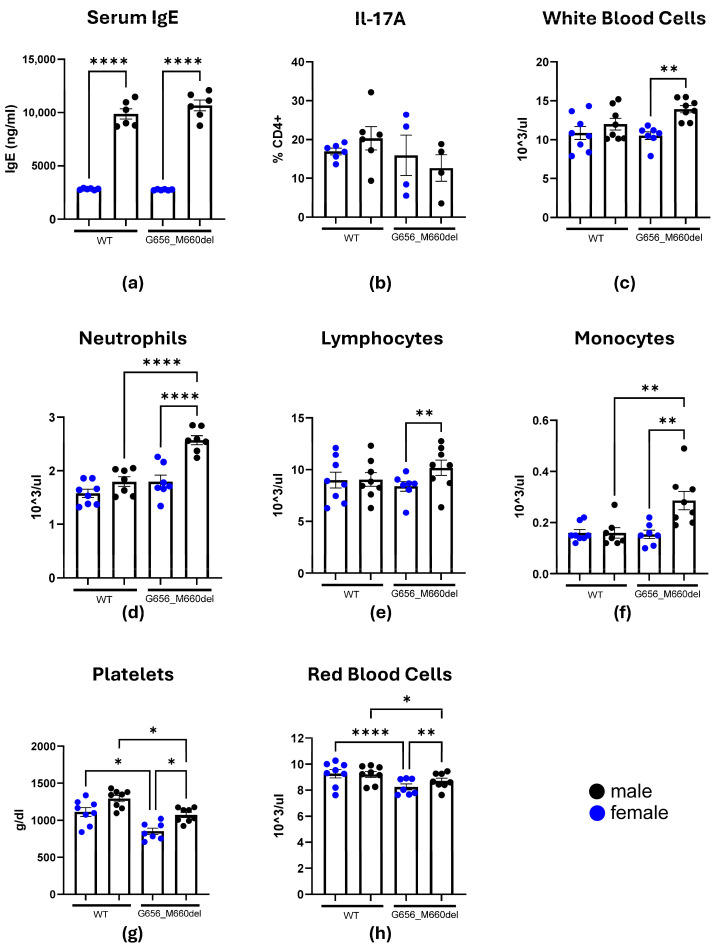
*G656_M660del* mice are characterized by a number of changes in hematopoietic population, but do not have AD-HIES phenotype. (**a**) Serum IgE as measured by ELISA, *n* = 6; (**b**) IL-17A-expressing CD4 T-cells in peripheral blood measured by flow cytometry. (**c**–**h**) Complete blood count results for measures significantly different between experimental groups. *n* = 7–8 * *p* < 0.05; ** *p* < 0.01; **** *p* < 0.0001; bar = SEM.

**Figure 2 ijms-26-07675-f002:**
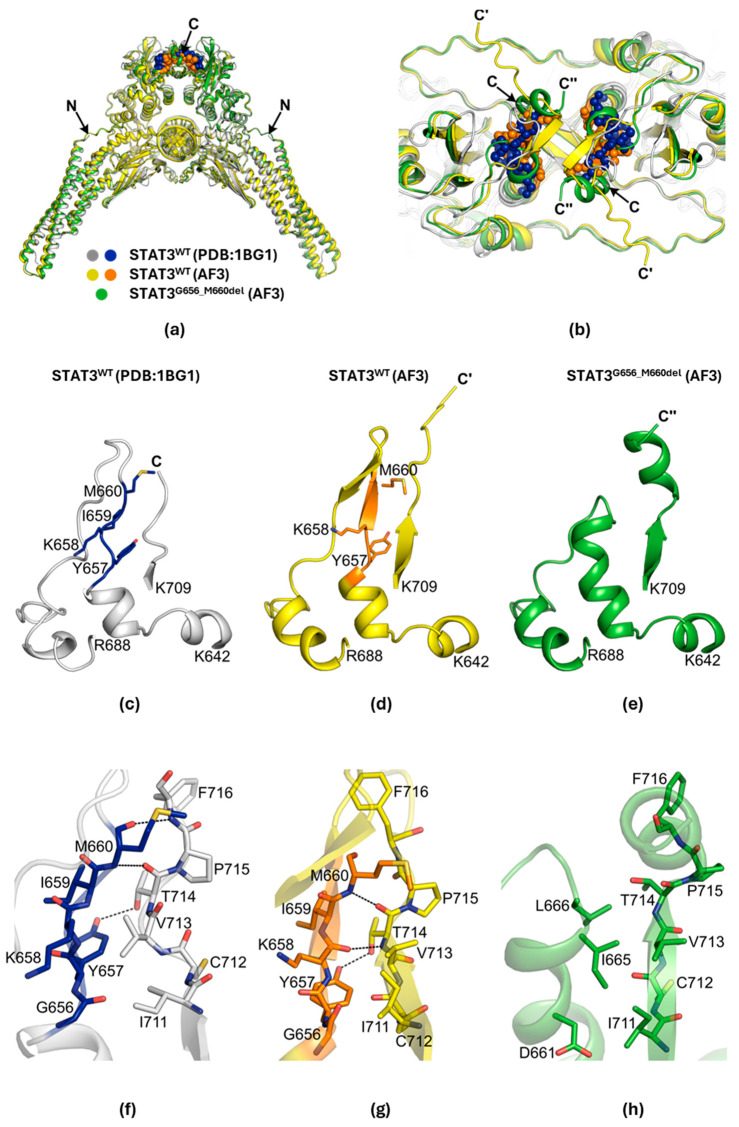
Structural comparison of STAT3 WT and *G656_M660del* mutant. (**a**) Overlay of the crystal structure of STAT3 WT dimer bound to dsDNA (gray and blue; PDB: 1BG1) withAF3-predicted WT (yellow and orange) and 656-660 deletion mutant (green, STAT3^G656_M660del^) structures. Proteins and dsDNA are shown in cartoon representation. The pentapeptide (^656^GYKIM^660^) is highlighted in blue (WT crystal) and orange (WT AF3). (**b**) Top view of the SH2 domain from panel (**a**), showing the relative positions of the C-termini: C (WT crystal), C′ (WT AF3), and C″ (mutant AF3). (**c**–**e**) Local structures surrounding the 656-660 region in (**c**) WT crystal, (**d**) WT AF3, and (**e**) mutant AF3 models. (**f**–**h**) Hydrogen-bonding interactions between the α-helix and the C-terminal β-strand for (**f**) WT crystal, (**g**) WT AF3, and (**h**) mutant AF3 structures. Hydrogen bonds are indicated by black dashed lines.

## Data Availability

Additional data, including raw CBC readouts, flow cytometry gating and results, as well as AlphaFold3 output are available in the Zenodo repository under the same title and DOI:10.5281/zenodo.16746906 (latest update at time of publication: version v2, 5 August 2025). Additional details are available on request from the correspondence author.

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
