# Peer review of "Mouse Model of STAT3 Mutation Resulting in Job’s Syndrome Diverges from Human Pathology"

_ijms, 2025, doi:10.3390/ijms26167675_

Round 1
Reviewer 1 Report
Comments and Suggestions for Authors
The current manuscript by Jankowski et al. presents findings from a mouse model intended to mimic human HIES caused by a 15-nucleotide deletion in the SH2 domain of STAT3. The authors report sex-specific differences in immune cell frequencies but no elevation in IgE levels. They conclude that, while this model may not be particularly useful for studying hyper-IgE syndrome, it can still serve as a valuable tool for fundamental research on STAT3 biology.Although the study lacks immediate clinical relevance, it is well-executed given the limited material available and provides important insights into STAT3 function in mice. Notably, it highlights fundamental differences in immune system regulation between humans and mice. The authors also include structural modelling of the described mutations, offering a useful resource for those interested in further exploration.While it would be interesting to see functional experiments demonstrating STAT3 activity in immune cells from this model, I believe the report already contains enough valuable information as is.
Minor comment:
Please ensure that all figure headings are formatted consistently. For example, in Figure 1, "serum IgE", "white blood cells", and "neutrophils" appear on different lines.
Reviewer 2 Report
Comments and Suggestions for Authors
Peer Review of “Mouse Model of STAT3 Mutation Resulting in Job’s Syndrome Diverges From Human Pathology” by Jankowski et al.
Decision: Major revision
Summary
Authors describe a heterozygous STAT3^G656_M660del mouse model corresponding to a human AD-HIES variant. Despite comprehensive CRISPR genotyping (Fig. S1) and longitudinal analyses of blood counts and immune subsets (Figs. 1, S2–S4), mice do not develop elevated IgE or eosinophilia. Instead, they show sex-specific alterations in leukocyte, erythrocyte, and platelet numbers. AlphaFold3 predicts structural perturbations in the SH2 domain and a modified environment around Y705, suggesting potential impacts on phosphorylation and dimerization.
- Major Points
• Lack of biochemical validation of Y705 phosphorylation
– Structural models predict loss of key hydrogen bonds and helix extension near Y705.
– No experimental data confirm altered Y705 phosphorylation or nuclear translocation.
– Recommended assay: IL-6/IL-10 stimulation of primary splenocytes or bone-marrow cells followed by Western blot for pY705-STAT3 and immunofluorescence or subcellular fractionation.
• Absence of direct readouts for STAT3 transcriptional activity
– Hematological shifts imply gain- or loss-of-function but remain indirect.
– Canonical STAT3 targets (e.g., Socs3, Bcl2, Ccl2) are not measured.
– Recommended assay: qRT-PCR for these targets in unstimulated and cytokine-stimulated cells to establish functional consequences.
• Incomplete immunophenotyping
– AD-HIES is characterized by Th17 deficiency and germinal-center B-cell defects, which are not addressed.
– Recommended additions: flow cytometry for CD4^+ IL-17A^+ (Th17), Foxp3^+ (Treg), GL7^+ Fas^+ (GC-B) subsets, and naïve versus memory T/B compartments.
• Genetic specificity concerns due to off-target CRISPR event
– The five-amino-acid deletion arose unintentionally, raising the possibility of confounding off-target mutations.
– Recommended approach: targeted sequencing of top predicted off-target sites or ≥2 generations of backcrossing to eliminate background variants.
• Statistical power and sex × genotype interaction
– Current sample sizes (n=6 WT, n=4 mutant per sex) limit the robustness of interaction analysis.
– Recommended action: expand cohorts to ≥8–10 animals per group, apply two-way ANOVA with interaction terms, and report exact p-values and effect sizes.
Minor Points
– Standardize variant nomenclature as G656_M660del throughout the text.
– Replace “dams”/“bucks” with “female”/“male.”
– Overlay individual data points on bar graphs in Figures 1 and S2–S4.
– Provide detailed flow-cytometry gating strategies in supplementary materials.
– Specify backcross generation number and exact mouse ages in Materials & Methods.
– Ensure uniform reference formatting (journal abbreviations, DOIs).
Recommendation
The manuscript presents valuable descriptive data and structural predictions, but requires additional mechanistic and genetic specificity experiments to substantiate the functional impact of the STAT3 deletion. Upon incorporation of the recommended assays and clarifications, the study will significantly advance understanding of STAT3 mutation phenotypes.
Reviewer 3 Report
Comments and Suggestions for Authors
This manuscript describes the method to design an animal model the mouse carrying mutations on the STAT3 gene, the same mutations present in patients with hyper-IgE syndrome (AD-HIES). They produced a model with STAT3G656_M660 that I understood would help understand the mechanisms of AD-HIES, i.e. it increases immune system cells but not IgE. All in all, after reading this manuscript a dozen times, that sure the methods are innovative and suitable to produce a mutated protein but honestly in this manuscript it is not clear or rather it is very entangled the purpose and the conclusions. It reads like random research.
Round 2
Reviewer 2 Report
Comments and Suggestions for Authors< !--StartFragment -->
Decision: Accept pending minor revisions
If the editors opt for a minor-revision decision, they will most likely ask you to:
- Verify all figure, table and supplement citations
- Ensure every “Figure Sx” and “Table S1” is actually referenced in the main text
- Check numbering and legends for consistency
2. Expand your statistical methods
- Specify which normality (e.g. Shapiro–Wilk) and variance (e.g. Levene’s) tests you ran
- For each ANOVA, note post-hoc corrections
3. Streamline the abstract
- Condense overlapping sentences on clinical features of HIES
- Highlight the novel findings in a single, punchy final statement
4. Standardize references
- Include full page ranges and DOIs for every citation
- Use a uniform author-year format for AlphaFold and related papers
5. Confirm data availability
- Test that your Zenodo DOI (10.5281/zenodo.15610072) resolves correctly
- Add the date of upload and version number, if applicable
6. Polish English usage
- Vary common phrases (“significant change,” “statistically significant”) with synonyms or passive constructions
- Eliminate mixed tenses in Methods and Results
Addressing these six points will satisfy a “minor revision” request and clear the way for final acceptance.
< !--EndFragment -->
Reviewer 3 Report
Comments and Suggestions for Authors
Thank you for your response, I understand your purpose.
